Calcitonin receptor expression in medullary thyroid carcinoma

Cappagli Virginia 1 2
http://orcid.org/0000-0002-5493-981X Potes Catarina Soares 3 4 5
http://orcid.org/0000-0001-5466-3027 Ferreira Luciana Bueno 1 3 6
Tavares Catarina 1 3 6
Eloy Catarina 1 3
Elisei Rossella 2
http://orcid.org/0000-0003-1613-1235 Sobrinho-Simões Manuel 1 3 7 8
http://orcid.org/0000-0002-3937-1621 Wookey Peter J. 9
http://orcid.org/0000-0001-9607-6998 Soares Paula 1 3 6 8 psoares@ipatimup.pt
1 Cancer Signaling and Metabolism Group, Institute of Molecular Pathology and Immunology of the University of Porto (IPATIMUP) , Porto , Portugal
2 Department of Clinical and Experimental Medicine, Endocrine Unit, University of Pisa , Pisa , Italy
3 Instituto de Investigação e Inovação em Saúde, Universidade do Porto , Porto , Portugal
4 Institute for Molecular and Cell Biology (IBMC), University of Porto , Porto , Portugal
5 Department of Biomedicine – Experimental Biology Unit, Faculty of Medicine, University of Porto , Porto , Portugal
6 Medical Faculty, University of Porto , Porto , Portugal
7 Department of Pathology, Hospital de S. João , Porto , Portugal
8 Department of Pathology, Medical Faculty, University of Porto , Porto , Portugal
9 Department of Medicine at Austin Health, University of Melbourne , Heidelberg, VIC , Australia
Petrie Kevin
Electronic publication date: 2017 Sep 13
Publication date: 2017
Volume: 5
Electronic Location ID: e3778
Received 2017 Jan 25; Accepted 2017 Aug 17
Copyright: © 2017 Cappagli et al.
Copyright year: 2017
Copyright holder: Cappagli et al.
License: This is an open access article distributed under the terms of the Creative Commons Attribution License, which permits unrestricted use, distribution, reproduction and adaptation in any medium and for any purpose provided that it is properly attributed. For attribution, the original author(s), title, publication source (PeerJ) and either DOI or URL of the article must be cited.
License URL: https://creativecommons.org/licenses/by/4.0/

Keywords: Calcitonin receptor, Calcitonin, C-cells, Medullary thyroid carcinoma

Funding: Fundação para a Ciência e Tecnologia SFRH/BPD/87537/2012 and SFRH/BD/87887/2012 National Counsel of Technological and Scientific Development, Brazil NORTE-01-0145-FEDER-000029 This study was supported by ERASMUS+ grant to VC and by Fundação para a Ciência e Tecnologia through the post-doctoral grant SFRH/BPD/87537/2012 to CSP and the doctoral grant SFRH/BD/87887/2012 for CT. This study was supported by a CNPq PhD Scholarship (“National Counsel of Technological and Scientific Development”, Brazil), Science Without Borders, Process n# 237322/2012-9 for LBF. Further funding from the project “Advancing cancer research: from basic knowledgment to application”; NORTE-01-0145-FEDER-000029; “Projetos Estruturados de I&D&I”, funded by Norte 2020 – Programa Operacional Regional do Norte; IPATIMUP integrates the i3S Research Unit, which is partially supported by FCT. This work is funded by FEDER funds through the Operational Programme for Competitiveness Factors – COMPETE and National Funds through the FCT, under the projects “PEst-C/SAU/LA0003/2013”. The funders had no role in study design, data collection and analysis, decision to publish, or preparation of the manuscript.

==============================
Background

Calcitonin expression is a well-established marker for medullary thyroid carcinoma (MTC); yet the role of calcitonin receptor (CTR), its seven-transmembrane G-protein coupled receptor, remains to be established in C-cells derived thyroid tumors. The aim of this work was to investigate CTR expression in MTC and to correlate such expression with clinicopathological features in order to evaluate its possible role as a prognostic indicator of disease aggressiveness and outcome.

Methods

Calcitonin receptor expression was analyzed in a series of 75 MTCs by immunohistochemistry, and by qPCR mRNA quantification in specimens from four patients. Statistical tests were used to evaluate the correlation between CTR expression and the clinicopathological and molecular characteristics of patients and tumors.

Results

Calcitonin receptor expression was detected in 62 out of 75 samples (82.7%), whereas 13 of the 75 samples (17.3%) were completely negative. CTR expression was significantly associated with expression of cytoplasmatic phosphatase and tensin homologue deleted on chromosome 10 and osteopontin, as well as with wild type RET/RAS genes and absence of tumor stroma, suggesting that CTR expression do not associate with clinicopathological signs of worse prognosis.

Discussion

Calcitonin receptor expression appears to be associated in MTC with more differentiated status of the neoplastic cells.

Introduction

Medullary thyroid carcinoma (MTC) is a rare tumor representing 5–10% of thyroid cancers (Elisei et al., 2013). It is a tumor with neuroendocrine differentiation and arises from the parafollicular C-cells of the thyroid which normally secrete calcitonin (CT). MTC originates as a sporadic (75–80%) malignancy or a manifestation of hereditary syndromes (20–25%), i.e., multiple endocrine neoplasia type 2 (MEN2A or MEN2B)/familial MTC, with an autosomal dominant pattern due to germline mutations of the RET gene (Pacini et al., 2010). In both forms of MTC (sporadic and familial) the clinic-laboratorial diagnosis is based mainly on the finding of elevated levels of serum CT, in basal and stimulated conditions. Serum CT is a very sensitive and specific method for diagnosing MTC, even though some other pathological/physiological conditions can be associated with increased levels of that hormone (Elisei et al., 2013). The clinical behavior of MTC is less favorable compared to follicular cell-derived thyroid tumors: 10 year survival rate is about 50–75% and the most important prognostic factor is tumor stage at diagnosis (Elisei et al., 2013; Kloos et al., 2009).

Calcitonin is a polypeptide hormone of 32 amino acids which is involved in the regulation of calcium homeostasis (Muff et al., 2004; Wimalawansa, 1997) under conditions of hypercalcaemia (Turner et al., 2011). It has been shown the osteoanabolic action of CT (Keller et al., 2014). CT has also been implicated in protecting the skeleton from excessive loss of bone during periods of high calcium demand, such as lactation (Davey & Findlay, 2013). In experimental models CT secretion was inhibited by high levels of CT suggesting a possible negative feedback mechanism (as for other endocrine system molecules) and autocrine regulation of normal C-cells (Kakudo et al., 1989; Morimoto et al., 1984; Orme & Pento, 1976).

Calcitonin binds specific calcitonin receptors (CTRs) that belong to the family B of G-protein coupled receptors (Chakraborty et al., 1991; Conner et al., 2002; Lin et al., 1991; Poyner et al., 2002). In mammals, CTR is widely expressed during blastula implant and during fetal (Jagger, Chambers & Pondel, 2000; Pondel, 2000; Tolcos et al., 2003) and perinatal (Tikellis et al., 2003; Wookey, Turner & Furness, 2012b) development. In adult tissues, CTR is widely expressed, for example in neural networks (Becskei et al., 2004; Sexton, McKenzie & Mendelsohn, 1988), in osteoclasts and osteocytes (Gooi et al., 2010), renal distal epithelium, B and T-cells (Body et al., 1990; Cafforio et al., 2009), testis (Chausmer, Stuart & Stevens, 1980), lung (Fouchereau-Peron et al., 1981) and several other tissues (reviewed in Findlay (2006) and Wookey et al. (2010)). CTR is also expressed by specific cell types in wound healing (Wookey et al., 2010), in cardiovascular diseases (Wookey et al., 2008; Wookey, Zulli & Hare, 2009) and in several types of malignant tissues as breast (Gillespie et al., 1997) and prostate cancer (Thomas et al., 2006), as well as in cell lines derived from neoplasias of the lung (Findlay et al., 1980, Findlay, Michelangeli & Robinson, 1989), breast (Findlay et al., 1981; Gillespie et al., 1997; Kuestner et al., 1994), brain (Wookey et al., 2012a), bone osteoclasts (Gorn et al., 1995; Nicholson et al., 1987), prostate (Thomas, Muralidharan & Shah, 2007), and of lymphoid (Marx et al., 1974) and myeloid tissues (Gattei et al., 1992; Silvestris et al., 2008).

Calcitonin receptor is the only receptor for CT characterized to date, and serves also as the signaling protomer for the heteromeric amylin receptor (CTR/RAMP-1). A unifying physiological role for CTR and its ligands in the previously mentioned situations remains to be advanced.

Calcitonin receptor function is best characterized as coupling to the stimulatory Gα subunit to increase adenylate cyclase (cAMP) activity and to activate downstream cAMP sensors PKA and Epac but has also been shown to couple to intracellular calcium mobilization and extracellular regulated kinase phosphorylation. In humans, two different isoforms of CTR, generated by alternative splicing, have been reported. These two forms differ by an insert of 16 amino acids in the first intracellular loop (CTR C1b [I+, insert+], 483 amino acids while CTR C1a [I−, insert−] is 467 amino acids in length) (Frendo et al., 1994; Gorn et al., 1992; Kuestner et al., 1994).

Frendo et al. demonstrated for the first time in 1994, the expression of the isoform CTR1a in thyroid cell lines (TT cell line derived from MTC) and in two MTC cases (Frendo et al., 1994). In subsequent studies Frendo et al. showed that CTR C1a mRNA was present in both normal and tumoral MTC thyroid tissue. No differences were found between sporadic and familial MTC regarding the expression of CTR C1a mRNA (Frendo et al., 1998a, 1998b). Higher expression of CTR C1a mRNA was found in MTC samples compared with normal tissue, and it was also reported that CTR C1a mRNA levels were modified during cell proliferation (Frendo et al., 1998b). The putative function of CTR in malignant tissues and tumor cell lines are still largely unknown and different results have been reported according to the different affected tissue. In breast cancer cell lines, there is evidence of an anti-proliferative effect of CTR activation (Nakamura et al., 2007; Ng et al., 1983), while in prostate cancer the up-regulation of the CT/CTR axis seems to help the switch of prostate cells towards a malignant phenotype (Thomas et al., 2006, Thomas, Muralidharan & Shah, 2007) stimulating proliferation, metastization and angiogenesis.

There are no studies evaluating levels of expression of CTR protein on MTC. Furthermore, no association between CTR mRNA or serum CT levels with the clinical characteristics or the prognosis of the patients with MTC have been reported (Frendo et al., 1998b). Given the lack of data regarding protein expression and the possible role of CTR in MTC, we decided to evaluate the expression of CTR protein in a large series of MTC and to correlate the expression level with molecular and clinicopathological features.

Materials and Methods

Human MTC tissue samples

A total of 75 MTC samples diagnosed in two institutions were used in the present study. Formalin-fixed, paraffin-embedded tissue and the corresponding clinical data were retrieved from the files of the Centro Hospitalar S. João (CHSJ)/Medical Faculty of Porto (FMUP)/Ipatimup (55 cases) and the Portuguese Institute of Oncology, Coimbra (IPO-C) (20 cases). The diagnosis of MTC was revised by two pathologists (CE and MSS) and confirmed by calcitonin immunostaining. Clinicopathological and follow-up data were obtained from the surgical pathology reports and patients’ records of the Department of Pathology and Oncology of CHSJ and from IPO database (Supplementary Table). The series from Ipatimup included consultation cases from which only limited demographic and clinical information was available. RET and RAS genetic characterization of the series have been previously reported (Lyra et al., 2014). The study was approved by the Hospital Ethical Committee of the Centro Hospitalar São João/Faculdade de Medicina da Universidade do Porto (CES 284/13) and the National Ethical rules were followed in every procedure.

Immunohistochemistry

Immunohistochemistry (IHC) for human CTR was performed in representative tumor sections of the 75 MTC cases. The mouse monoclonal anti-human CTR antibody (mAb) 31/01-1H10 against a cytoplasmatic epitope within the carboxyl terminal of human CTR (DIPIYICHQELRNEPANN; Welcome Receptor Antibodies Pty Ltd., Melbourne, Australia; also distributed as MCA2191 by BioRad AbD Serotec) was used, which was already characterized in previous studies (Silvestris et al., 2008; Wookey et al., 2008, 2012a; Wookey, Zulli & Hare, 2009). Deparaffinized and rehydrated sections were subjected to microwave treatment in 10 mM sodium citrate buffer, pH 6.0, for antigen retrieval. After blocking, the sections were incubated overnight at 4 °C in a humidified chamber with the primary antibody anti-CTR (mAb 31/01-1H10 1:4,000). For the detection, a labelled streptavidin–biotin immunoperoxidase detection system was employed (Thermo Scientific/Lab Vision, Fremont, CA, USA), and the immunohistochemical staining was developed with 3,3′-diaminobenzidine substrate. A negative control consisting on the omission of the primary antibody was performed. IHC evaluation was performed independently by two observers (CE and VC). CTR expression was evaluated taking into account the proportion of stained cells (scored as ≤5% = 0; 5–25% = 1; 25–50% = 2, 50–75% = 3 and 75–100% = 4) and the staining intensity (scored as absent = 0, faint = 1, moderate = 2 and strong = 3) (Table 1). CTR expression was semi-quantified using a staining score (from 0 to 12) corresponding to the multiplication of the staining intensity by the proportion of positive stained cells (Table 2) as previously described by our group (Ferreira et al., 2016; Lyra et al., 2014). CTR expression was correlated with data previously obtained by our group in this series of tumors with regard to phosphatase and tensin homologue deleted on chromosome 10 (PTEN), phospho-S6 ribosomal protein (pS6) (Lyra et al., 2014) and osteopontin (OPN) IHC expression (Ferreira et al., 2016).

Table 1 Staining intensity and extension of CTR expression in the 75 MTC cases.

Intensity expression	n	%	Cellular expression (%)	n	%	
Absent	11	14.7	<5	13	17.3	
Faint	26	34.7	5–25	0	0	
Moderate	19	25.3	25–50	7	9.3	
Strong	19	25.3	50–75	4	5.3	
			75–100	51	68	
Total	75	100		75	100	

Table 2 Staining score of CTR IHC in the 75 MTC cases.

CTR staining scorea	n	%	
0	13	17.3	
2	6	8	
3	4	5.3	
4	15	20	
8	18	24	
12	19	25.3	
Total	75	100%	
Note:

a Product of the staining intensity by the proportion of positive cells; scores of 1, 5, 6, 7, 9, 10 and 11 were not obtained in any case.

Calcitonin staining with a rabbit monoclonal antibody (ref.: RM-9117-S, clone SP17, Neomarkers) was performed for diagnostic purposes and CT immune-expression was semi-quantified, based on the intensity of the staining, in a score from 1 to 4. In the majority of the cases the staining of CT and CTR was done in serial sections. The CT score obtained was correlated with the corresponding CTR expression score in each case (64/75 cases; in 11 cases we do not have access to the calcitonin staining slides).

RNA extraction and reverse transcription

Total RNA was extracted from frozen specimens of MTC (n = 4), from adjacent normal tissue specimens (n = 5) and from two MTC-derived cell lines (TT, purchased from American Type Culture Collection – ATCC; and MZ-CRC-1, provided by Dr. Robert Hofstra, Netherlands) using a Trizol commercial kit (Life Technologies; GIBCO BRL, Carlsbad, CA, USA) according to the manufacturer’s protocol. RNA was quantified by spectrophotometry and its quality was checked by analysis of 260/280 nm and 260/230 nm ratios. For cDNA preparation, 1 μg of total RNA was reverse transcribed using the RevertAid first strand cDNA synthesis kit (Fermentas, Burlington, ON, Canada).

Real time PCR

Reverse transcription products of CTR were amplified by real-time quantitative PCR (#HS.PT.56a.40988589; IDT, Coralville, IA, USA) using the TaqMan® PCR Master Mix (Applied Biosystems, Foster City, CA, USA) with TBP gene (TATA-binding protein) as endogenous control (#4326322E-0705006; Applied Biosystems). The ABI PRISM 7500 Fast Sequence Detection System (Applied Biosystems) was used to detect the amplification level and was programmed to an initial step of 2 min at 50 °C, 10 min at 95 °C, followed by 45 cycles of 95 °C for 15 s and 60 °C for 1 min. The relative quantification of target genes was determined using the ΔΔCT method, which was previously validated by Livak’s linear regression method (slope = 0.0696) (Sequence Detector User Bulletin 2; Applied Biosystems). Primers used for qPCR are available at the manufacturer’s website.

Statistical analysis

Statistical analysis was performed using 22.0 SPSS statistical package (IBM, Armonk, NY, USA). The relationship between the immunohistochemical score of CTR and clinicopathological features was evaluated by independent sample t-test or Mann–Whitney test (for comparisons of groups having less than 30 cases). The correlation between the immunoreactivity of the other proteins (PTEN, pS6, OPN and CT) with CTR was assessed using the Pearson correlation test. A p ≤ 0.05 was considered statistically significant.

Results

CTR protein expression in MTC

Calcitonin receptor expression was mainly localized in the cytoplasm and was detected in 62 out of 75 samples (82.7%), while the remaining 13 samples (17.3%) were negative. In the 62 positive samples, CTR expression was present in more than 50% of the cells in 55 cases (88.7%) (Table 1) and the staining intensity was faint in 34.7%, moderate in 25.3% and strong in 25.3% (Table 1). The distribution of the staining score among the positive cases is present in Table 2. In a few cases, scattered nuclear staining was observed. Examples of representative cases and the negative controls are shown in Fig. 1 (additional staining patterns are shown in Fig. S1).

Figure 1 CTR expression in MTC.

(A and B) negative control, a MTC case in which the primary antibody was omitted. (C and D) negative CTR-expression in a case of MTC. (E and F) positive CTR-expression in a case of MTC (score 2: extent 25–50%, intensity 1+). (G and H) positive CTR-expression in a case of MTC (score 12: extent 100%, intensity 3+). The dashed square in (A), (B) and (C) photomicrographs taken at 10× magnification represents the area in (D), (E) and (F) pictures taken using the 60× objective. Bar 100 μm.

Correlation of CTR expression with clinicopathological and molecular features of MTC

No significant associations were observed between CTR expression and age of the patients, tumor dimension, lymph node and/or distant metastases tumoral invasion nor extrathyroidal extension (Table 3). Tumors from female patients had significantly higher CTR expression (Table 3).

Table 3 Clinicopathological and molecular associations with CTR expression.

Clinicopathological (N)	CTR expression (mean ± SD)	pc Value	
Gender		0.044	
 Female (29)	(6.8 ± 4.8)	
 Male (25)	(4.4 ± 3.50)	
Tumor size (cm)		0.55	
 <2 (18)	(5.7 ± 4.6)	
 ≥2 (20)	(4.9 ± 4.5)	
Stroma		0.042	
 Absent (20)	(7.9 ± 4.2)	
 Present (hyaline) (51)	(5.6 ± 4.3)	
Amyloid deposits		0.91	
 Absent (14)	(5.5 ± 4.2)	
 Present (26)	(5.4 ± 4.6)	
Extrathyroidal extensiona		0.42	
 Absent (7)	9.5	
 Present (14)	11.8	
Metastasesa		0.96	
 Absent (11)	15.1	
 Present (18)	14.9	
Invasion (vascular and/or capsular)a		0.62	
 Absent (4)	14.6	
 Present (21)	12.7	
RET		0.26	
 Wild type (37)	(6.65 ± 4.54)	
 Mutated (38)	(5.53 ± 4.19)	
RAS		0.11	
 Wild type (66)	(6.38 ± 4.37)	
 Mutated (9)	(3.89 ± 3.88)	
RET or RAS mutation		0.046	
 Wild type (29)	(7.34 ± 4.47)	
 Mutated (46)b	(5.28 ± 4.16)	
Notes:

a For these variables the Mann–Whitney non parametric test (group < 30 cases) was used; the CTR protein expression is reported as mean rank.

b A case presented a RET and a RAS mutation.

c The bold entries correspond to statistical significant values.

The characteristics of the tumoral stroma were evaluated separately, accounting for the presence of either an amyloid stroma or a hyaline/desmoplastic stroma versus the absence of any kind of stroma. MTC samples without desmoplastic stroma had significantly higher expression of CTR (7.9 vs 5.5, p = 0.04) than tumors presenting hyaline/desmoplastic stroma, while there was no correlation with the presence/absence of an amyloid stroma.

Tumors wild-type for RAS or RET genes had higher CTR expression when compared to mutated cases (7.3 vs 5.2, p = 0.04; Table 2). The same trend was observed when these mutations were analyzed separately but the differences did not reach statistical significance (p = 0.07, for RAS-positive cases versus wild-type for both mutations and p = 0.09 for RET-positive tumors versus tumors wild-type for both mutations).

There was a strong positive correlation between CT staining and CTR expression, (p = 0.001), that is, the cases with higher score for CT staining showed also higher CTR expression (Fig. 2).

Figure 2 CT and CTR staining in serial MTC tissue sections.

(A and D) CTR score 0 and CT intensity 1. (B and E) CTR score 4 (extent 75–100%, intensity 1), CT intensity 3. (C and F) CTR score 12 (extent 75–100%, intensity 3), CT intensity 4. Note that higher CTR scores correspond to more intense CT stainings. Photomicrographs were taken at 10× magnification. Bar 100 μm.

The immunohistochemical detection of other cancer related proteins such as PTEN, pS6 and OPN had been previously analyzed in the same series of MTCs (Lyra et al., 2014; Ferreira et al., 2016). In this study we evaluated the relationship between the expression of those markers with CTR expression. Tumors with cytoplasmatic PTEN expression presented significantly higher CTR expression compared to PTEN cytoplasmatic negative cases (p = 0.038). There was also a significant association between OPN expression and CTR expression (p = 0.009). No correlation was found between CTR and pS6 expression (p = 0.21).

CTR mRNA expression in MTC

Calcitonin receptor mRNA levels were analyzed in four cases of MTC and adjacent non-tumoral thyroid tissue from which frozen samples were available, and also in two MTC-derived cell lines (TT and MZ-CRC-1). A similar expression of CTR mRNA was observed in adjacent thyroid tissue and MTC (Fig. 1) except in one case in which a much higher expression was observed in the tumor than in the respective adjacent thyroid parenchyma. We must emphasize that the low number of C-cells present in adjacent normal parenchyma limits this analysis. Both MTC-derived cell lines expressed CTR mRNA (Fig. 3); the expression was higher in TT cell line than in MZ-CRC-1 (TT = 0.498658 vs MZ-CRC-1 = 0.1280699).

Figure 3 CTR mRNA expression in thyroid tissues and in two MTC cell lines.

Discussion

This is the first study in which expression levels of CTR protein were evaluated in a large series of MTC cases. The study of the expression of CTR mRNA and of the function of CT/CTR in MTC is limited to few papers that analyzed a limited number of cases (Frendo et al., 1998a, 1998b, 1994). In the present study the expression of CTR protein was evaluated in 75 MTC cases and this information was used to search for correlations with clinicopathological and molecular features. We observed CTR expression in 82.7% (62/75) of MTC cases.

At variance with the results obtained in other tumor models, in our series high levels of CTR protein expression did not correlate with poor prognosis or aggressive features of MTC. On the contrary, there was a tendency for CTR to be more expressed in smaller tumors, without capsular or vascular invasion and without distant metastases. These findings could suggest that CTR expression might be associated to tumor differentiation. Further studies in large series are necessary to confirm (or not) this tendency.

Frendo et al. (1998a, 1994) observed that in MTC and TT cell lines, the shorter isoform of CTR (CTR C1a) was expressed and that CTR mRNA was present both in normal and tumor tissue with higher levels in tumors and without differences among the different clinical forms of MTC (Frendo et al., 1998b). Our hypothesis that higher expression of CTR in MTC may be associated with a more differentiated status of the neoplastic cells is consistent with the demonstration that inhibition of proliferation of the TT cell line correlated with increased expression and secretion of CT (deBustros et al., 1986). The expression of CTR mRNA in two MTC derived cell lines open the possibility of further functional studies with siRNA to evaluate in vitro the consequences of CTR gene silencing in the differentiation of MTC cells. The close homology between several molecules: CT/CTR (independent of RAMPs), amylin receptors with CTR/RAMPs 1, 2 or 3 and, CGRP receptors defined by CTR/RAMP 2 (as well as CRLR/RAMP 1) can, however, be a limiting factor in those studies.

We must also consider the possibility of transient epithelial–mesenchymal transition (EMT) as it was recently reported by Johansson et al. (2015) in MTC where “differentiation genes” are repressed in locally invasive tumor cells but re-expressed at metastatic sites. In this sense it will be also very interesting to explore the expression of CTR in primary and metastatic MTC lesions, as well as in the invasive front vs tumor bulk, in combination with lineage specific markers (Foxa1/Foxa2) and epithelial–mesenchymal transition markers.

In our hands, CTR expression correlated with CT expression at the protein (IHC) level. Taken together, these data fit with the assumption that a more differentiated MTC status, as evaluated by CT levels, is associated with elevated expression of CTR (Fig. 2).

There seems to be an association of OPN expression and a more differentiated MTC status. Briese et al. (2010) demonstrated a higher expression of the OPN protein in MTC than in normal thyroid. Our group confirmed this association through the finding that OPN is correlated with features of better prognosis of MTC and with C-cell differentiation (Ferreira et al., 2016). OPN is a matricellular glycoprotein involved in biological processes, as biomineralization, bone remodeling and immune responses, and also in pathological processes. In the present study we found a strong positive correlation between CTR expression and OPN expression and this finding reinforces the hypothesis that CTR is probably related with C-cell differentiation.

Phosphatase and tensin homologue deleted on chromosome 10 is a phosphatase enzyme and acts as tumor suppressor with different functions according to its subcellular localization (Bononi & Pinton, 2015; Chung & Eng, 2005). PTEN is an important down-regulator of Akt/mTOR, a pathway that is involved in MTC tumorigenesis (Tamburrino et al., 2012). In a previous study of our group (Lyra et al., 2014) we proposed that, in MTCs, RAS mutation plays a direct role in the activation of mTOR pathway, while in RAS wild type tumors the mTOR pathway appear to be activated by a mechanism involving a lower expression of cytoplasmatic PTEN. In the present study we observed that higher expression of CTR correlated with higher cytoplasmic PTEN expression in MTC. In accordance with our previous results we found that tumors wild-type for RAS or RET had significantly higher CTR expression when compared to mutated tumors. Further studies are necessary to evaluate the role of CTR in this context.

The role of stroma in the regulation of tumorigenesis is largely acknowledged (Tlsty & Coussens, 2006). The stromal reaction has been described in several tumor types and it has been related with a more invasive and aggressive tumor behavior in most of such models (Rowley, 1998). This holds particularly true in MTC in which several studies (Koperek et al., 2008; Scheuba et al., 2006) have associated the presence of a “hyaline/desmoplastic” stroma with more aggressive features (tumor size, tumor stage, more invasive tumors and the presence of lymph node metastases). In the present series, we observed a significantly higher expression of CTR in cases without or with discrete tumor stroma. Putting these results together we think that our results reinforce the idea that CTR may be a marker of better differentiation, less invasive and less aggressive tumors.

The CT/CTR axis in tumors is not well understood and the studies on record report dissimilar data according to different tumor models. In prostate, several studies (Chigurupati et al., 2005; Shah et al., 2009; Thakkar et al., 2013; Thomas et al., 2006, Thomas, Muralidharan & Shah, 2007), using primary tumor samples and cell lines, suggested a tumorigenic role of CTR. In particular, it was demonstrated that there was a different spatial expression between normal and tumor tissue with a higher expression in the tumors and that, while in the early stage of prostate cancer the tumor cells expressed either CT or CTR, in the advanced cases there was a co-expression of both and such co-expression associated with a metastatic phenotype (Thakkar et al., 2013). Furthermore, a higher expression of CT/CTR correlated with a higher tumor grade and a worsen prognosis (Thakkar et al., 2013). In prostate cancer, the most important pathway upregulated through CTR activation is cAMP, leading to a higher invasiveness due to the degradation of extracellular matrix by PKA and urokinase-plasminogen A system (Thomas et al., 2006). In contrast to this, in breast cancer, CT/CTR seems to play a protective role as Ng et al. (1983) showed that CT was able to inhibit the growth of a breast cancer cell line. In subsequent studies, other groups (Gillespie et al., 1997; Wang et al., 2004) demonstrated that CTR, mostly the isoform 2, was expressed in tumor samples and cell lines and that a decreased CTR expression was observed in advanced tumors with lymph node metastases and lymphatic invasion. Finally, CTR was shown to be involved in the control of breast cancer invasion by downregulating the activity of urokinase-plasminogen A and inhibiting cells invasiveness in a concentration-dependent manner (Han et al., 2006). Our results regarding the CTR expression in MTC and its correlation with patients’ prognosis are more alike the breast cancer model, namely regarding the tendency for CTR to be more expressed in smaller tumors, without invasion and metastases. Unfortunately, data concerning the final outcome of the patients was only available in 20 out of the 75 cases and this represents a major limitation. From these 20 cases 13 were CTR-positive (65%) and 7 were CTR-negative (35%). Considering the outcome of the patients, seven out of 20 patients are free of disease being five CTR-positive (71%) and two CTR-negative (29%) whereas in the group of the five patients that died due to the disease, two (40%) were CTR-positive and three (60%) were CTR-negative. This tendency to lower CTR expression in patients with guarded prognosis needs to be verified in a larger MTC series.

Conclusion

In summary, the present study confirms that CTR is expressed in most MTCs and our data seems to suggest that CTR expression in MTCs is associated with a more differentiated status and clinical and molecular features of good prognosis. Further studies are needed to clarify the function of CTR in normal and tumoral C-cells.

Supplemental Information

Supplemental Information 1 Summary of the clinical, pathological and molecular data of the MTC cases.

Click here for additional data file.

Supplemental Information 2 Representative photomicrographs of each score of calcitonin receptor protein expression.

(A) Score 0 absence of staining. (B) Score 2 (extent 25–50%, intensity 1+). (C) Score 3 (extent 50–75%, intensity 1+). (D) Score 4 (extent 25–50%, intensity 2+). (E) Score 8 (extent 75–100%, intensity 2+). (F) (extent 100% intensity 3+). Photomic rographs were taken at 10x magnification. Bar 100μm.

Click here for additional data file.

Supplemental Information 3 Raw data.

Click here for additional data file.

Supplemental Information 4 Raw data PCR.

Click here for additional data file.

Additional Information and Declarations

Competing Interests

Author Contributions

Human Ethics

Data Availability

Paula Soares is an Academic Editor for PeerJ.

Virginia Cappagli performed the experiments, analyzed the data, wrote the paper, prepared figures and/or tables.

Catarina Soares Potes conceived and designed the experiments, performed the experiments, analyzed the data, wrote the paper, prepared figures and/or tables.

Luciana Bueno Ferreira performed the experiments, analyzed the data, prepared figures and/or tables.

Catarina Tavares analyzed the data, prepared figures and/or tables.

Catarina Eloy analyzed the data.

Rossella Elisei reviewed drafts of the paper.

Manuel Sobrinho-Simões reviewed drafts of the paper.

Peter J. Wookey contributed reagents/materials/analysis tools, reviewed drafts of the paper.

Paula Soares conceived and designed the experiments, analyzed the data, contributed reagents/materials/analysis tools, wrote the paper, reviewed drafts of the paper.

The following information was supplied relating to ethical approvals (i.e., approving body and any reference numbers):

The Comissão de Ética para a Saúde (CES) do Centro Hospitalar de São João (CHSJ)/Faculdade de Medicina da Universidade do Porto (FMUP) granted ethical approval for the study (CES 284/13).

The following information was supplied regarding data availability:

The raw data has been supplied as Supplemental Dataset Files.

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
