# Peer review of "Calcitonin receptor expression in medullary thyroid carcinoma"

_PeerJ, doi:10.7717/peerj.3778_

## Round 0.1 · original submission · Major Revisions

This manuscript addresses the urgent need for new biomarkers for the prognosis of MTC. However, reviewers have identified several areas that need to addressed before it can be accepted for publication:

1. The IHC scoring for CTR should be as comprehensive as PTEN. This issue needs to be addressed. It should be possible to present the original IHC images for all stainings as a supplemental file. This would address many comments from authors. Negative controls should be included - these should be shown in Figure 1.

2. Was IHC performed on serial sections? - this needs to stated and if not this needs to be discussed.

Minor points:

The authors could consider using quantitative in situ RNA expression assays such as RNAscope or Panomics. This not cheap or easy to perform and is not required for resubmission. It does, however, provide very useful contextual information.

In the discussion, the authors should address the lack of statistically significant association between CTR expression and clinical data and moderate their claims regarding prognostic relevance until further studies have been undertaken.

Reviewer 1 ·

Basic reporting

Please consider reformulating sentences in lines 61 – 64 for a better understanding.

Experimental design

No comment.

Validity of the findings

Results, clearly presented by Cappagli V et al, add information to current knowledge concerning molecular profile of MTC. A correlation between CTR expression and clinical outcome is not supported by data presented and might be avoided.

Additional comments

It is stated that CTR mRNA levels were analyzed in four cases of MTC and adjacent non-tumoral thyroid tissue conducting to the conclusion that a similar expression of CTR mRNA was observed in adjacent thyroid tissue and MTC. However, adjacent thyroid tissue is mainly derived from follicular cells. Does it mean that C-cells and follicular cells similarly express CTR?

The authors did not find significant associations between CTR expression and any of the following parameters: tumor size, stroma, amyloid deposits, extrathyroidal extension, metastases and vascular an/or capsular invasion.
Data regarding final outcome was only available in 20 out of the 75 cases. Among these cases, only 7 (35.5%) were CTR-positive contrasting with the whole group in which 82.7 were CTR-positive. Moreover, from the CTR-positive subgroup died 28.5% patients whereas from the CTR-negative subgroup died 23% patients.
All together it seems an overstatement to conclude that CTR expression appears to be associated with a better prognosis.

·

Basic reporting

Raw data for comparative analysis of co-expression of proteins (CTR vs CT, CTR vs PTEN, CT vs OPN) are lacking.

Experimental design

The rationale of investigating CTR expression in C cell-derived tumors that express the natural ligand (calcitonin), inferring an autocrine type of regulation, should be better explained and discussed. For instance, are there any information in the literature suggesting normal C cells themselves are regulated by CT? If such a mechanism is unknown it should be told so in the Introduction.

Method of data presentation on CTR expression in Table 2 should be explained. As judged from the Materials and Methods section it appears to be based on a scoring method combining IHC staining intensity and extent of staining. The general validity of this calculation may also be valuable to readers.

Validity of the findings

The discordance of overall IHC staining intensity and expression of CTR at the cellular level (cells seem either strongly positive or negative) , as presented in Table 1, should be commented on. Does this mean that faint/moderate immunostaining depends on fewer positive cells rather a lower expression level in the majority of cells? Figs. 3B and E argue in favor of the latter, so are these images indeed representative?

Imaging the expression of both CTR and CT in serial sections from the same tumor specimens is warranted to support the statement in Results of a statistically significant correlation with p<0.001. In fact, the same criticism accounts for CTR vs PTEN and CRT vs OPN. Cases (inter- or intratumoral) with differential expression pattern (CTRhigh but no CT/PTEN/OPN, or the reverse), if present, would also be of interest to report.

Additional comments

It is suggested that in their discussion of tumor differentation/dedifferentiation authors also consider the possibility of transient epithelial-mesenchymal transition (EMT), in which "differentiation genes" are repressed in locally invasive tumor cells (and circulating cancer cells) but re-expressed at metastatic sites. That this accounts for human MTC was recently reported (Johansson et al, Development 142: 3519-28, 2015). It is thus expected that CT and possibly also CTR are down-regulated in MTC cells that undergo EMT, which may explain some of the present findings.

Identification of high CRT mRNA levels in MTC cell lines in this study opens for functional studies with siRNA to eludicate a possible autocrine pathway. This may be mentioned in the paper.

Distinction of CT/CTC and CGRP/CGRP-receptors may be of interest to clarify to readers.

Reviewer 3 ·

Basic reporting

Basic reporting is acceptable with sufficient references , good quality photomicrographs and structured article format with clear english language.
However ,
1. Introduction is quite detailed and lengthy. The authors may consider revising the introduction so that it is short and to the point.
2. Raw data does not provide sufficient detail on CTR immunohistochemistry expression findings (column P and Q of raw data table), unlike rest of the markers where detail findings of cellular expression and intensity have been mentioned. Kindly explain the headings in P and Q column of raw data table: CTR+ and CTRx ?
3.If RET positive cases were 38 and RAS were 9, kindly explain RET+RAS mutation is 46 in Table 2.

Experimental design

Research question, investigation and methodology are basically well defined. Experimental design is relevant in the medical and pathological context and within the scope of peerj i.e. – Medical sciences. Investigation is rigorous.

Validity of the findings

1. Authors may consider combining the effects of cellular expression and intensity to give an idea of effect on overall for CTR expression in Table 1.
2. As the authors have stated, there is no statistically significant association between CTR expression and clinical data such as tumour size etc. The authors may consider revising the discussion and conclusion accordingly.

Additional comments

Overall, the study is informative and relevant in the current scientific context. Few revisions need to be made before acceptance.

---

## Round 0.2 · accepted · Accept

The authors have addressed the reviewers' points comprehensively and the manuscript is much improved.

Reviewer 1 ·

Basic reporting

No comment.

Experimental design

No comment.

Validity of the findings

No comment.